# UNFAIR GEOMETRIES: EXACTLY SOLVABLE DATA MODEL WITH FAIRNESS IMPLICATIONS

## ABSTRACT

Machine learning (ML) may be oblivious to human bias but it is not immune to its perpetuation. Marginalisation and iniquitous group representation are often traceable in the very data used for training, and may be reflected or even enhanced by the learning models. In the present work, we aim at clarifying the role played by data geometry in the emergence of ML bias. We introduce an exactly solvable high-dimensional model of data imbalance, where parametric control over the many bias-inducing factors allows for an extensive exploration of the bias inheritance mechanism. Through the tools of statistical physics, we analytically characterise the typical properties of learning models trained in this synthetic framework and obtain exact predictions for the observables that are commonly employed for fairness assessment. Despite the simplicity of the data model, we retrace and unpack typical unfairness behaviour observed on real-world datasets. We also obtain a detailed analytical characterisation of a class of bias mitigation strategies. We first consider a basic loss-reweighing scheme, which allows for an implicit minimisation of different unfairness metrics, and quantify the incompatibilities between some existing fairness criteria. Then, we consider a novel mitigation strategy based on a matched inference approach, consisting in the introduction of coupled learning models. Our theoretical analysis of this approach shows that the coupled strategy can strike superior fairness-accuracy trade-offs.

## 1 INTRODUCTION

Machine Learning (ML) systems are actively being integrated in multiple aspects of our lives, from face recognition systems on our phones, to applications in the fashion industry, to high stake scenarios like healthcare. Together with the advantages of automatising these processes, however, we must also face the consequences of their — often hidden — failures. Recent studies Buolamwini & Gebru (2018); Weidinger et al. (2021) have shown that these systems may have significant disparity in failure rates across the multiple sub-populations targeted in the application. ML systems appear to perpetuate discriminatory behaviours that align with those present in our society Benjamin (2019); Noble (2018); Eubanks (2018); Broussard (2018). Discrimination over marginalised groups could originate at many levels in the ML pipeline, from the very problem definition, to data collection, to the training and deployment of the ML algorithm Suresh & Guttag (2021).

Data represents a critical source of bias Perez (2019). In some cases, the dataset can contain a record of a history of discriminatory behaviour, causing complex dependencies that are hardly eradicated even when the explicit discriminatory attribute is removed. In other cases (or even concurrently), the root of the discrimination can be found in the data collection process, and is related to the structural properties of the dataset. Heterogeneous representations of different sub-populations typically induce major bias in the ML predictions. Drug testing provides a historically significant example: substantial evidence Hughes (2007); Perez (2019) shows that the scarcity of data points corresponding to women individuals in drug-efficiency studies resulted in a larger number of side effects in their group.

In spite of a vast empirical literature, a large gap remains in the theoretical understanding of the bias-induction mechanism. A better theoretical grasp of this issue could help raise awareness and design more theoretically grounded and effective solutions. In this work, we aim to address this gap by introducing a novel synthetic data model, offering a controlled setting where data imbalances and the emergence of bias become more transparent and can be better understood.

To the best of our knowledge, the present study constitutes the first attempt to explore and exactly characterise by analytical means the complex phenomenology of ML fairness.

**Summary of main results.**    We devise a novel synthetic model of data, the *Teacher-Mixture* (T-M), to obtain a theoretical analysis of the bias-induction mechanism. The geometrical properties of the model are motivated by common observations on the data structure in realistic datasets, concerning the coexistence of non-trivial correlations at the level of the inputs and between inputs and labels (some empirical observations can be found in appendix B). In particular, we focus on the role played by the presence of different sub-populations in the data, both from the point of view of the input distribution and from that of the labelling rule. Surprisingly, this simple structural feature is sufficient for producing a rich and realistic ML fairness phenomenology.

The parameters of the T-M can be tuned to emulate disparate learning regimes, allowing for an exploration of the impact of each bias-inducing factor and for an assessment of the effectiveness of a tractable class of mitigation strategies. In summary, in the present work we:

- Derive, through a statistical physics approach, an analytical characterisation of the typical performance of solutions of the T-M problem in the high-dimensional limit. The obtained learning curves are found to be in perfect agreement with numerical simulations in the same synthetic settings (as shown in the central panel in Fig. 1), and produce unfairness behaviours that are closely reminiscent of the results seen on real data.

- Isolate the different sources of bias (shown in the left panel of Fig. 1) and evaluate their interplay in the bias-induction mechanism. This analysis also allows us to highlight how unfairness can emerge in settings where the data distribution is apparently balanced.

- Trace a positive transfer effect between the different sub-populations, which implies that, despite their distinctions, an overall similarity can be exploited for achieving better performance on each group.

- Analyse the trade-offs between the different definitions of fairness, by studying the effects of a sample reweighing mitigation strategy, which can be encompassed in the theoretical framework proposed in this work and thus characterised analytically.

- Propose a model-matched mitigation strategy, where two coupled networks are simultaneously trained and can specialise on different sub-populations while mutually transferring useful information. We analytically characterise its effectiveness, finding that with this method, in the T-M, the competition between accuracy and different fairness metrics becomes negligible. Preliminary positive results are also reported on real data.

**Further related works.**    In the past decade, algorithmic fairness has been receiving growing attention, spurred by the increasing number of ML applications in highly consequential social and economic areas Datta et al. (2015); Metz & Satariano (2020); Angwin et al. (2016). A central question in the field is on the proper mathematical definition of bias: the plethora of alternative fairness criteria includes measures of *group fairness*, e.g. statistical parity Corbett-Davies et al. (2017); Dwork et al. (2012); Kleinberg et al. (2016), disparate impact Calders & Verwer (2010); Feldman et al. (2015); Zafar et al. (2017b); Chouldechova (2017), equality of opportunity Hardt et al. (2016), calibration within groups Kleinberg et al. (2016), disparate mistreatment Zafar et al. (2017a), as well as measures of *individual fairness* Speicher et al. (2018); Castelnovo et al. (2022). We focus on group fairness in the following, since it is well-defined also in the high-dimensional limit considered in our theoretical framework. Recent works have highlighted incompatibilities between some of these fairness measures Kleinberg et al. (2016); Corbett-Davies & Goel (2018); Barocas et al. (2019), e.g. calibration and error disparity Pleiss et al. (2017), and their instability with respect to fluctuations in the training dataset Friedler et al. (2019); Castelnovo et al. (2022). Our work is the first to allow an exact quantification of the intrinsic trade-offs between these notions of group-fairness.

A second major topic in the field of algorithmic fairness is that of bias mitigation. In this work, we focus on *in-processing* strategies Arrieta et al. (2020), where the training process is altered in order to include fairness as a secondary optimisation objective for the learning model. These methods range from including *ad hoc* regularisation terms to the loss function Kamishima et al. (2012); Huang & Vishnoi (2019), to formulating fair classification as a constrained optimisation problem and deriving reduction-based algorithms Agarwal et al. (2018; 2019); Celis et al. (2019). Other possible strategies include adversarial training Zhang et al. (2018), where a fairness-arbiter model can drive learning towards a sough fairness criterion, and distributionally robust optimisation Słowik & Bottou (2021),

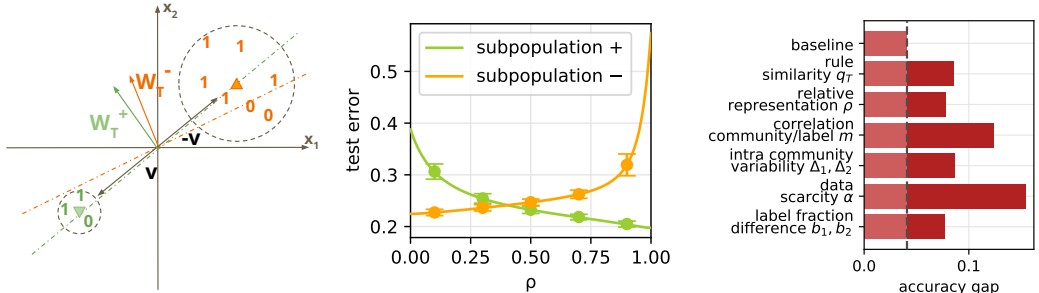

Figure 1: **T-M model**. Often we can distinguish sub-populations as clusters in a dataset according to some features. The label, e.g. effectiveness of a drug, is given by some rule acting on the data an may differ for the two subpopulations. In the T-M model, *(Left)*, the two sub-populations are drawn from two Gaussians around the two centres (green and orange triangles). The labels (plus and minus) are associated according to the hyper-planes $W_T^+, W_T^-$. In this 2D drawing we can see that the group $+$ (green) has 3 samples while group $-$ (orange) has 7 samples, so $\rho = 0.3$. The two hyper-planes are highly overlapping ($q_T \approx 1$) and weakly aligned with the shift vector ($m_+ \approx 0$, $m_- \approx 0$). Finally, we see that sub-population $+$ is less spread than sub-population $- (\Delta_+ < \Delta_-)$. *(Centre)* For this model, the test error can be calculated exactly, shown here as a function of $\rho$ (solid curve). Numerical simulations (dots) closely match the analytical results. The panel exemplify the importance of $\rho$ in creating bias against one sub-population or the other. *(Right)* Effect of changing one of the model parameter in terms of test accuracy gap, starting the from the set-up of the central panel with $\rho = 0.2$.

where one accounts for worst-case unfairness scenarios across the sub-populations in the data. In this work, we analyse two simple schemes whose performance can be analytically traced in our framework. First, an approach Kamiran & Calders (2012); Plecko & Meinshausen (2020); Lum & Johndrow (2016) based on loss-reweighing according to the associated subgroup and label of each data point. Second, we propose –and analyse– a novel method based on the introduction of coupled learning models, which can be interpreted as a modification of the "two naive Bayes" model in Calders & Verwer (2010).

Alternative classes of debiasing approaches, which cannot be analysed within our framework, include pre-processing strategies Calmon et al. (2017); Feldman et al. (2015), learning unbiased representations Zemel et al. (2013), and post-processing techniques based on Decision Theory and Causal Reasoning Kamiran et al. (2012); Plecko & Meinshausen (2020).

## 2 MODELLING DATA IMBALANCE

The Teacher-Mixture model, sketched in Fig. 1, combines aspects of two common modelling frameworks for supervised learning, namely the Gaussian-Mixture (GM) and the Teacher-Student (TS) setups. The GM is a simple model of clustered input data, where each data point is sampled from one of a small set of –possibly overlapping– high-dimensional Gaussian distributions, while the TS provides a simple model of input-label correlation, where the ground-truth labels are obtained from a random "teacher" neural network and the "student" learning model tries to reproduce similar outputs. While retaining analytical tractability, the novel T-M data model allows for a richer phenomenology than the previous models, retracing the main features of real data with multiple coexisting sub-populations. For simplicity, the results discussed in this paper will focus on the case of two groups, but the analysis could be extended to multiple sub-populations.

More formally, we consider a synthetic dataset of $n$ samples $\mathcal{D} = \{\mathbf{x}^\mu, y^\mu\}_{\mu=1}^n$, with $\mathbf{x}^\mu \in \mathbb{R}^d$, $y^\mu \in \{0, 1\}$. We define the $\mathcal{O}(1)$ ratio $\alpha = n/d$ and we refer to it as the data scarsity parameter. Each input vector is i.i.d. sampled from a mixture of two symmetric Gaussians with variances $\Delta = \{\Delta_+, \Delta_-\}$, $\mathbf{x} \sim \mathcal{N}(\pm\boldsymbol{v}/\sqrt{d}, \Delta_\pm\mathbb{I}^{d\times d})$, with respective probabilities $\rho$ and $(1-\rho)$. The shift vector $\mathbf{v}$ is a Gaussian vector with i.i.d. entries with zero mean and variance 1. The $1/\sqrt{d}$ scaling corresponds to the *high-noise* noise regime, where the two Gaussian clouds are overlapping and hard to disentangle Mignacco et al. (2020); Saglietti & Zdeborová (2022), e.g. as in the case of

CelebA and MEPS shown in appendix C. The ground-truth labels, instead, are provided by two i.i.d. Gaussian teacher vectors, namely $\mathbf{W}_T^+$ and $\mathbf{W}_T^-$, with components of zero mean and variance 1. Each teacher produces labels for the inputs with the corresponding group-membership, namely $y^\mu = \text{sign}\left(\mathbf{W}_T^\pm \cdot \mathbf{x}_\pm^\mu + \mathbf{b}_T^\pm\right)$. The thresholds $\mathbf{b}_T^\pm$ correspond to the teacher bias terms, included in the model to control the fraction of positive and negative samples within the two sub-populations. Overall, the geometric picture of the data distribution (a sketch in Fig. 1) is summarised by the following overlaps:

$$\tilde{m}_\pm = \frac{1}{d}\boldsymbol{W}_T^\pm \cdot \boldsymbol{v} \qquad q_T = \frac{1}{d}\boldsymbol{W}_T^+ \cdot \boldsymbol{W}_T^-, \tag{1}$$

that respectively quantify the alignment of the teacher decision boundaries with respect to the shift vector, controlling the group-label correlation, and the overlap between the teacher vectors, controlling the correlation between labels assigned to similar inputs belonging to different communities.

Given the synthetic dataset $\mathcal{D}$, we study the properties of a single-layer network $\boldsymbol{W}$ trained via empirical risk minimization (ERM) of the loss:

$$\mathcal{L}(\boldsymbol{w}) = \sum_{\mu \in \mathcal{D}} \ell\left(\frac{\boldsymbol{W}_T^{c^\mu} \cdot \boldsymbol{x}^\mu}{\sqrt{d}} + \tilde{b}_{c^\mu}, \frac{\boldsymbol{W} \cdot \boldsymbol{x}^\mu}{\sqrt{d}} + b_s\right) + \frac{\lambda}{2}\left(\sum_{i=1}^d w_i^2\right) \tag{2}$$

where $\ell(y, \hat{y})$ is assumed to be convex, $\lambda$ is an external parameter that regulates the intensity of the $L_2$ regularisation, and the index $c^\mu \in \{+, -\}$ denotes the group membership of data point $\mu$.

Given this framework, we derive a theoretical characterisation of the asymptotics of this learning model and consider the possible implications from a ML fairness perspective. In particular, we aim at studying the role of data geometry and cardinality in the training of a fair classifier. To quantify the level of bias in the predictions of the trained model, we need to choose a metric of fairness. We will diffusely employ *disparate impact* (DI) Feldman et al. (2015), a ML analogous of the 80% rule Commission et al. (1979), which allows a simple assessment of the over-specialisation of the classifier on one of the sub-populations. In principle, in the T-M there is no preferable realisation of the target attribute so we can adopt a symmetric version of $DI = p(\hat{y} = y|+)/p(\hat{y} = y|-)$, defined as the ratio between test accuracy in sub-population $+$ and sub-population $-$. How to measure bias is itself an active line of research and the DI measure is imperfect. In Sec. 4 we compare with other metrics.

Note that the T-M has, at the same time, the advantage of being simple, allowing better understanding of the many facets of ML bias, and the disadvantage of being simple, since some modelling assumptions might not reflect the complexity of real-world data. For example, we ignore any type of correlation among the inputs other than the clustering structure. The goal of this modelling work continues a long tradition of research in statistical physics, which has shown that theoretical insights gained in prototypical settings can often be helpful to disentangle and interpret the complexity of real world behaviour.

**Remark 1** *By looking at the available degrees of freedom in the T-M, several possible sources of bias naturally emerge from the model:*

- *the* relative representation, $\rho = n_+/(n_+ + n_-)$, *with $n_c$ the number of points in group c.*

- *the* group variance, $\Delta_c$, *determining the width of the clusters.*

- *the* label frequencies, *controlled through the bias terms $\mathbf{b}_c$.*

- *the* group-label correlation, $m_c$.

- *the* labelling rule similarity, $q_T$, *which measures the alignment between the two teachers, i.e. the linear discriminators that assign the labels to the two groups of inputs.*

- *the* data scarcity, $\alpha$, *representing the ratio between dataset size and input dimension.*

**Theoretical analysis in high-dimensions.** In principle, solving Eq. 2 requires finding the minimiser of a complex non-linear, high-dimensional, quenched random function. Fortunately, statistical physics Mézard et al. (1987) showed that in the limit $n, d \to \infty, n/d = \alpha$, a large class of problems, including the T-M model, becomes analytically tractable. In fact, in this proportional high-dimensional

regime, the behaviour of the learning model becomes deterministic and trackable due to the strong concentration properties of a narrow set of descriptors that specify the relevant geometrical properties of the ERM estimator. The original high-dimensional learning problem can be reduced to a simple system of equations that depends on a set of scalar overlaps:

$$Q = \frac{1}{d}\boldsymbol{W} \cdot \boldsymbol{W}, \qquad m = \frac{1}{d}\boldsymbol{W} \cdot \boldsymbol{v} \qquad R_\pm = \frac{1}{d}\boldsymbol{W} \cdot \boldsymbol{W}_T^\pm, \qquad (3)$$

representing the typical norm of the trained estimator, its magnetisation in the direction of the cluster centres, and its overlap with the two teachers of the T-M.

**Analytical result 1** *In the high dimensional limit when $n, d \to \infty$ at a fixed ratio $\alpha = n/d$, the scalar descriptors $\Theta = \{Q, m, R_\pm, \delta q\}$ of the vector $\boldsymbol{w}$ obtained by the empirical risk minimisation of Eq. 2 with a convex loss, and their Lagrange multipliers $\hat{\Theta} = \{\hat{Q}, \hat{m}, \hat{R}_\pm, \delta\hat{q}\}$, converge to deterministic quantities given by the unique fixed point of the system:*

$$Q = -2\frac{\partial s(\hat{\Theta}; \lambda)}{\partial\, \delta\hat{q}}; \quad m = \frac{\partial s(\hat{\Theta}; \lambda)}{\partial\, \hat{m}}; \quad R_\pm = \frac{\partial s(\hat{\Theta}; \lambda)}{\partial\, \hat{R}_\pm}; \quad \delta q = 2\frac{\partial s(\hat{\Theta}; \lambda)}{\partial\, \hat{Q}}; \qquad (4)$$

$$\hat{Q} = 2\alpha\frac{\partial e(\Theta; \Delta)}{\partial\, \delta q}; \quad \hat{m} = \alpha\frac{\partial e(\Theta; \Delta)}{\partial\, m}; \quad \hat{R}_\pm = \alpha\frac{\partial e(\Theta; \Delta_\pm)}{\partial\, R_\pm}; \quad \delta\hat{q} = 2\alpha\frac{\partial e(\Theta; \Delta)}{\partial\, Q}; \qquad (5)$$

*with:*

$$s(\hat{\Theta}; \lambda) = \frac{\hat{Q} + \left(\hat{m} + \sum_{c=\pm} \tilde{m}_c \hat{R}_c\right)^2 + \sum_{c=\pm}(1 - \tilde{m}_c^2)\hat{R}_c^2 + 2\left(q_T - \prod_{c=\pm} \tilde{m}_c\right)\prod_{c=\pm} \hat{R}_c}{2\,(\delta\hat{q} + \lambda)} \qquad (6)$$

$$e(\Theta; \Delta) = \mathbb{E}_c\left[\mathbb{E}_z \sum_{y=\pm1} H\left(-y\frac{\sqrt{Q}(c\,\tilde{m}_c + \tilde{b}_c) + \sqrt{\Delta_c}R_c z}{\sqrt{\Delta_c(Q - R_c^2)}}\right) v(y, c, \Theta)\right] \qquad (7)$$

*where $c \in \{+, -\} \sim \text{Bernoulli}(\rho)$, $z \sim \mathcal{N}(0, 1)$, $H(\cdot) = \frac{1}{2}\text{erfc}(\cdot/\sqrt{2})$ is the Gaussian tail function, $w$ is the solution of:*

$$v(y, c, \Theta) = \max_w\left[-\frac{w^2}{2} - \ell\left(y, \sqrt{\Delta_c\delta q}w + \sqrt{\Delta_c Q}z + c\,m + b\right)\right] \qquad (8)$$

*and the bias $b$ implicitly solves the equation $\partial_b e(\Theta; \Delta) = 0$.*

Note that this result was obtained through the non rigorous yet exact replica method from statistical physics Mézard et al. (1987); Engel & Van den Broeck (2001); Zdeborová & Krzakala (2016). The derivation details are deferred to appendix D. We remark that several analytic results obtained through the replica method have been subsequently proved rigorously. In particular, the proofs presented by Thrampoulidis et al. (2015); Mignacco et al. (2020); Loureiro et al. (2021) in settings similar to the present one suggest that an extension for the T-M case could be derived. However, this is left for future work. In this manuscript, we verify the validity of our theory by comparison with numerical simulations, as shown e.g. in the central panel of Fig. 1.

The obtained fixed point for the scalar descriptors $\Theta$ can be used to evaluate simple expressions for common model evaluation metrics, such as the *confusion matrix* or the *generalisation error*.

**Analytical result 2** *In the same limit as in Analytical result 1, the entries of the confusion matrix, representing the probability of classifying as $\hat{y}$ an instance sampled from sub-population $c$ with true label $y$, are given by:*

$$p(\hat{y} \,|\, y; c) = \mathbb{E}_z\left[\text{Heav}\left(y\left(\sqrt{\Delta_c}z + c\,\tilde{m}_c + \tilde{b}_c\right)\right) H\left(-\hat{y}\frac{(c\,m + b) + \sqrt{\Delta_c}R_c z}{\sqrt{\Delta_c(Q - R_c^2)}}\right)\right], \qquad (9)$$

*where $z \sim \mathcal{N}(0, 1)$ and $\text{Heav}(\cdot)$ is the Heaviside step function. The generalization error, representing the fraction of wrongly labelled instances, can then be obtained as $\epsilon_g = \mathbb{E}_c\left[\sum_{\hat{y}\neq y} p(\hat{y} \,|\, y; c)\right]$.*

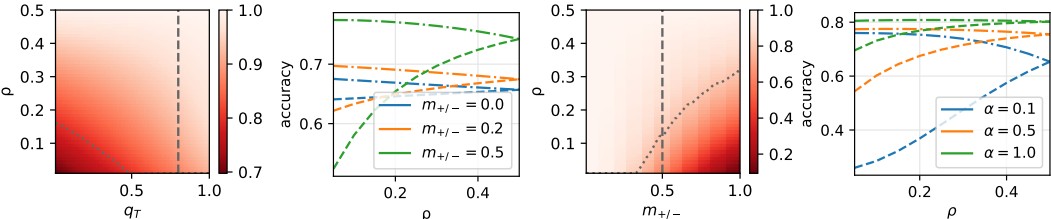

Figure 2: **Bias under different parametric settings.** Impact of several parameters on the Disparate Impact (DI) of the model. From left to right. *(Panel 1)* Phase diagram where each point represents the DI (red indicates a worse accuracy on group $+$) for different values of rule similarity $q_T$ (x-axis) and the relative representation $\rho$ (y-axis). The dotted grey line denotes the 80% threshold for disparate impact. *(Panel 2)* Accuracy of group $+$ (dashed lines) and group $-$ (dot-dashed lines), in a cut across the first phase diagram at $q_T = 0.8$. The different colours indicate different levels of group-label correlation $m_\pm$. *(Panel 3)* Phase diagram of the DI at fixed $q_T = 1$, as the group-label correlation $m_\pm$ (x-axis) and $\rho$ are varied. *(Panel 4)* Role of the dataset size ($\alpha$), at a cut $m_\pm = 0.5$ of the diagram in panel 3.

This second result provides us with a fully deterministic estimate of the accuracy of the trained model on the different data sub-populations. These scores will be used in the following sections to investigate the possible presence of bias in the classification output of the model. Note that theorems 1 and 2 allow for an extremely efficient and exact evaluation of the learning performance in the T-M, remapping the original high-dimensional optimisation problem onto a system of deterministic scalar equations that can be easily solved by recursion.

## 3 INVESTIGATING THE SOURCES OF BIAS

With these analytical results at hand, we now turn to systematically investigating the effect of the sources of bias identified in remark 1, which potentially mine the design of a fair classifier. We consider three separate experiments to summarise some distinctive features of the fairness behaviour in the T-M: namely, the impact of the correlation between the labelling rules and the group structure, the interplay between relative representation and group variance, and the positive transfer effect in the data-scarse regime. The parameters of the experiments, if not specified in the caption, are detailed in appendix E.1.

**Group-label correlation.** In the two left panels of Fig. 2, we consider a scenario where the labelling rules for the two groups are not perfectly aligned, i.e. $\boldsymbol{W}_T^+ \neq \boldsymbol{W}_T^-$ (and/or $b_+ \neq b_-$). Note that in this case we have a clear mismatch between the learning model, a single linear classifier, and the true input-output structure in the data: the learning model cannot reach perfect generalisation for both sub-populations at the same time. For simplicity, we set an equal correlation between the two teacher vectors and the shift vector, $m_+ = m_- > 0$, and isolate the role of rule similarity $q_T$. The first panel shows a phase diagram of the DI (DI$< 1$ indicating a lower accuracy on group $+$), as function of the similarity of the teachers and the fraction of $+$ samples in the dataset. As intuitively expected, the induced bias exceed the 80% rule when the labelling rules are misaligned and the group sizes are numerically unbalanced (small $q_T$ and $\rho$). Indeed, in the cut displayed in the second panel, by lowering the group-label correlation $m_\pm$ the gap between the measured accuracies on the two sub-populations becomes smaller. However:

**Remark 2** *Even when $q_T = 1$ and the task is solvable (i.e. the classifier can learn the input-output mapping), the trained model can still be biased.*

This is shown in the two panels on the right of Fig. 2, where a large high-bias region (DI$< 80\%$) exists. In particular, the third panel shows the cause of this effect in the presence of a non-zero group-label correlation $m_\pm$, and in the fourth panel we see how this effect is more pronounced in the data-scarse regime. In all four panels, as $\rho$ reaches $0.5$, the two sub-populations become equally represented and the classifier achieves the same accuracy for both.

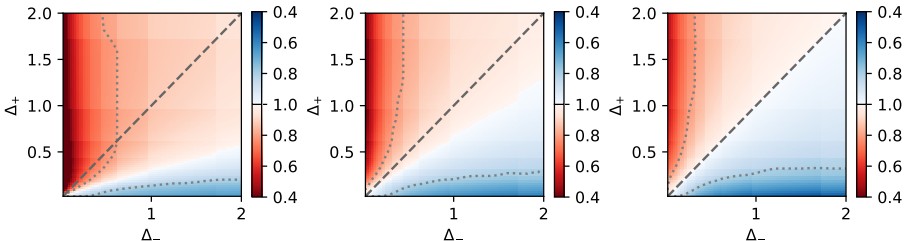

Figure 3: **Bias in equally represented subpopulations.** We show the disparate impact as the distribution of the two subpopulations is changed by altering their variances ($\Delta_+$ and $\Delta_-$). The diagonal line gives the configurations where the two subpopulations have the same variance. The three panels consider different levels of representation, from left to right $\rho = 0.1, 0.3, 0.5$. The latter is the situation with both subpopulations being equally represented in the dataset. We use the red and blue colours to quantify the disparate bias against sub-population $+$ and $-$ (respectively).

**Bias and variance.** In Fig. 3, we plot the DI as a function of the group variances $\Delta_\pm$, for different values of the fraction of $+$ samples. One finds that the model might need a disproportionate number of samples in the two groups to obtain comparable accuracies. We can see that:

**Remark 3** *Balancing the group relative representation does not guarantee a fair training outcome.*

In fact, the quality of a group's representation in the dataset can increase if the number of points is kept constant but the group variance is reduced. The blue regions in the first two panels indicate a higher accuracy for the minority group even if the dataset only contains $10\%$ and $30\%$ of samples belonging to it. This exemplifies the fact that a very focused distribution (low $\Delta_\pm$) actually requires less samples. The last panel ($\rho = 0.5$) shows the scenario one would expect *a priori*: on the diagonal line the DI is balanced, but by setting $\Delta_+ > \Delta_-$ (or viceversa) one induces a bias in the classification.

**Positive transfer.** If mixing different sub-populations in the same dataset can induce unfair behaviour, one could think of splitting the data and train independent models. In Fig. 4, we show that a *positive transfer effect* Gerace et al. (2022) can yet be traced between the two groups when the rules are sufficiently similar.

**Remark 4** *The performance on the smaller group tends to further deteriorate if the dataset is split according to the sub-group structure.*

To clarify this point, we plot the DI as a function the data scarcity $\alpha$, for several values of the rule similarity $q_T$ and at fixed $\rho$. We also compare the accuracies on each sub-population of a classifier trained on the full dataset and of a baseline classifier trained only on the respective data subsets ($+$ in the second figure, $-$ in the third). If the rules are sufficiently similar (large $q_T$), we can observe a positive transfer and using the dataset in his entirety leads to a performance and fairness improvement.

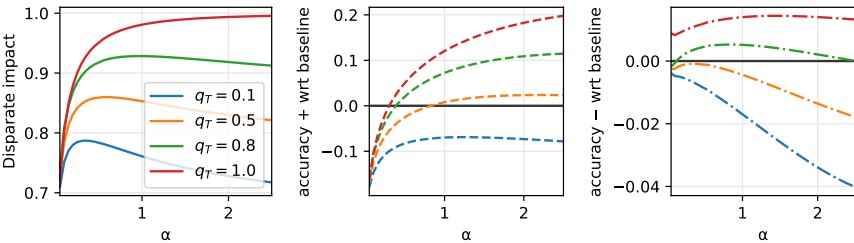

Figure 4: **Positive transfer effect.** Given a fixed proportion of the two sub-populations, we compare different levels of rule similarity ($q_T$) as the size of the dataset is increased. The disparate impact (first figure) may mislead into thinking that the accuracy in one sub-population is decreasing as the other increases, instead the accuracy is steadily increasing (second figure) for both sub-populations. Finally, the last two figures show the accuracy in the sub-population $+$ and $-$ (respectively) minus the accuracy on the same dataset when the other sub-population is perfectly removed.

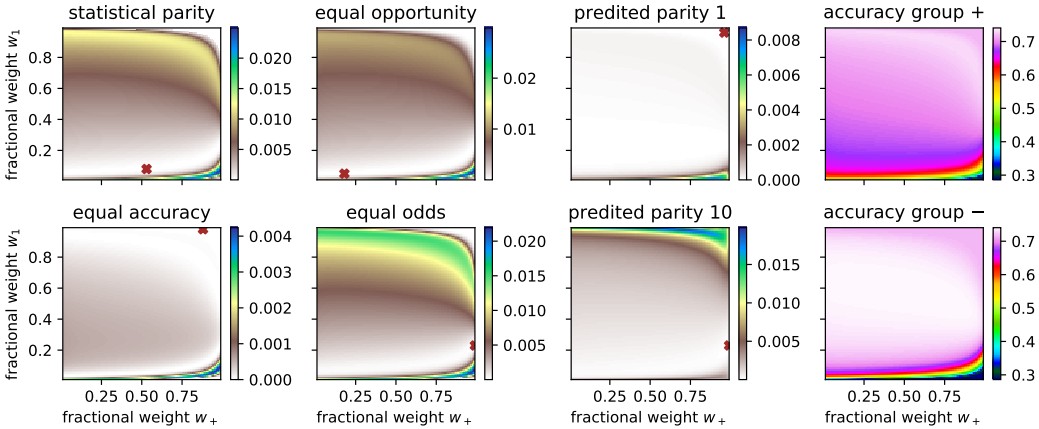

Figure 5: **Mitigation trade-off.** The phase diagrams show the effect of re-weighting, biasing both: towards low mistakes in classifying sub-population $+$ ($w_+$ on the x-axis) and towards low mistakes for label $+1$ ($w_1$ on the y-axis). The quantity shown in the diagrams are the mutual information for the metrics introduced in the text (first three columns) and the accuracy on the two sub-populations (last column). The diagrams for the mutual information also show red markers denoting where the minimum is achieved.

As expected, positive transfer can be particularly useful in data-scarce regimes (small $\alpha$) and becomes ineffective or detrimental in large datasets (large $\alpha$), as shown in the last panel.

## 4 MITIGATION STRATEGIES

To assess the fairness of a ML model on a given data distribution, a plethora of different fairness criteria have been designed Speicher et al. (2018); Castelnovo et al. (2022). Appendix F presents a summary of the criteria considered in our analysis. Following the lines of Speicher et al. (2018), we aim to quantify exactly how far is a given trained model from meeting each of these criteria. Given a classification event $E$ –specified by the criterion– and the group membership $C$, a natural measure of their independence is provided by the Mutual Information (MI):

$$I(E;C) = D_{KL}(\mathbb{P}[E,C] \,|\, \mathbb{P}[E]\mathbb{P}[C]) = \mathbb{E}_{(E,C)} \log \frac{\mathbb{P}[E,C]}{\mathbb{P}[E]\mathbb{P}[C]}. \tag{10}$$

Clearly, the fairness condition is completely verified only if the joint distribution factorises, i.e. $\mathbb{P}[E,C] = \mathbb{P}[E]\mathbb{P}[C]$, and the mutual information goes to zero. This represents the impossibility of predicting the classification outcome of an unbiased model just from the group membership.

In the following, we consider two simple bias mitigation strategies that can be analysed within our analytical framework. The required generalisations of the replica results are detailed in appendix D. First, we study the de-biasing effect of a sample reweighing strategy where the relevance of each sample is varied based on its label and group membership Kamiran & Calders (2012); Plecko & Meinshausen (2020); Lum & Johndrow (2016). By adjusting the weights, one can indirectly minimise the MI relative to any given fairness measure. We use the simultaneous quantitative predictions on the various metrics to assess the compatibility between different fairness definitions. Then, we propose a theory-based mitigation protocol, along the lines of protocols used in the context of multi-task learning Rusu et al. (2016).

**Loss Reweighing.** Recent literature shows that some fairness constraints cannot be satisfied simultaneously. ML systems are instead forced to accept trade-offs between them Kleinberg et al. (2016). This sort of compromise is well-captured in the simple framework of the T-M model. Fig. 5 shows, in form of phase diagrams, the MI measured with respect to the various fairness criteria while varying the two reweighing parameters, $w_1$ and $w_+$, which up-weigh data points with true label $1$ and in group $+$, respectively. E.g., the loss term associated to a label $1$-group $+$ sample will be weighed $w_+ w_1$, while that of a label $0$-group $-$ data point will receive weight $(1 - w_+)(1 - w_1)$. By changing these relative weights one can force the model to pay more attention to some types of errors and re-establish a balance between the accuracies on the two sub-populations. The red

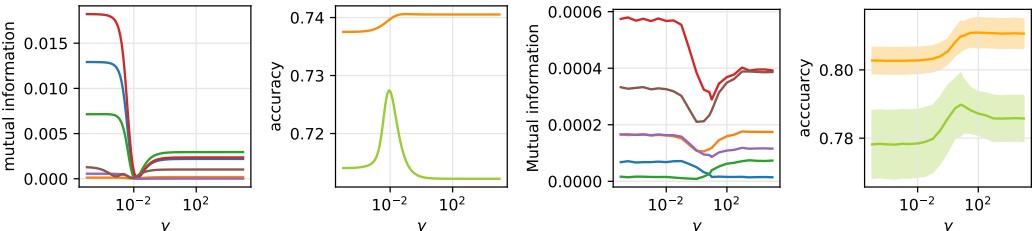

Figure 6: **Mitigation trade-off in the coupled architecture.** The first two figures represent a one dimensional version of Fig. 5 for the coupled architecture set up. On the left panel, the mutual information of the different fairness measures (statistical parity, equal opportunities, equal accuracy, equal odds, predicted parity 1, predicted parity 10) is plot as function of the coupling-strength parameter $\gamma$, observe that the minima of the curves are much closer. Furthermore, the second panel shows a better accuracy trade-off between subpopulation $+$ and subpopulation $-$. The remaining two figures, show an example from the CelebA dataset splitting and classifying according to the attributes "Wearing_Lipstick" and "Wavy_Hair" respectively, more details are provided in appendix B and C. The observations made for the synthetic model applies also in this real-world case.

crosses in the phase diagrams identify the points where the MI reaches its minimum value for each fairness metric. Notably, some minima are found to lie in different regions of the phase diagram (at the opposite extremes), and they often align only in correspondence of trivial classification, where fairness is achieved but at the expense of accuracy. These results are in agreement with rigorous results in the literature Barocas et al. (2019), but also show how the incompatibilities between the different constraints extend to regimes where the fairness criteria are not exactly satisfied.

**Coupled Networks.** The emergence of classification bias in the T-M could be lead back to the clear mismatch between the generative model of data and the learning model. In order to move towards a matched inference setting, we need to enhance the learning model to account for the presence of multiple sub-populations and labelling rules. This inspires a novel mitigation strategy – called *coupled neural networks*. The strategy consists in the simultaneous training of multiple neural networks, each one seeing a different subset of the data associated with a different sub-population. The networks exchange information by means of an elastic penalty that mutually attracts them, and the intensity of this elastic interaction is obtained by cross-validation. This approach is close in spirit to other methods already present in the literature Calders & Verwer (2010); Saglietti et al. (2021); Zenke et al. (2017).

**Remark 5** *The coupled neural networks method allows for higher expressivity and specialisation on the various sub-populations, while also encouraging a positive transfer between similarly labelled sub-groups, leading to better fairness-accuracy trade-offs*

The first plot in Fig. 6, displaying the behaviour of the mutual information as a function of the coupling parameter for different fairness metrics, shows the key advantage of using this method. We observe is a more robust consistency among the various fairness metrics: the positions of the different minima are now very close to each other. Moreover, the value of the coupling parameter achieving this agreement condition is also the one that minimises the gap in terms of test accuracy between the two sub-populations, as shown in the second plot of Fig. 6, without hindering the performance on the larger group. Notice that this result does not contradict the impossibility theorem Barocas et al. (2019) which states that statistical parity, equal odds, and predicted parity cannot be satisfied altogether. In fact, our result only concerns soft minimisation of each fairness metrics. In appendices D and F.1 we provide additional results for this method and we discuss the effect of training the networks on data subsets that only partially correlate with the true group structure.

Despite the fact that the T-M is just a data prototype, the positive agreement with real phenomenology suggests that this method could be effective also on real-world data. The remaining two plots in Fig. 6 show preliminary results of the performance of the coupled neural networks strategy in the realistic dataset from CelebA[1]. We stress that although the method works significantly better in the

---

[1]The illustrated chekpoints are used only to show the similarity of behavior in synthetic data and realistic data (CelebA), and not used or recommended to use in any face recognition systems or scenarios.

synthetic framework, real data present more complex correlations that may hinder the effectiveness of the method. Therefore, an application of this technique on real settings requires caution. A future research direction will be to understand the range of applicability of the coupled neural networks and, consequently, its limitations.

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
