# OpenReview forum: "Unfair geometries: exactly solvable data model with fairness implications"
_ICLR.cc/2023/Conference — Submitted to ICLR 2023_

### Official Review · Reviewer_NozC · 2022-10-24

**Confidence:** 2
**Correctness:** 3
**Technical Novelty And Significance:** 3
**Empirical Novelty And Significance:** 3
**Recommendation:** 5

**Clarity, Quality, Novelty And Reproducibility:**

This paper is not easy to read. There are too many results, some are consistent with existing works and some are new. The authors are encouraged to re-organize the results and conclude the results in each section.

**Strength And Weaknesses:**

Strength:

+. The framework of modeling data generation is simple and intuitive. The analytical results illustrate sources of bias.
+. The coupled neural networks method is interesting. The authors identified the classification bias in the T-M is due to the mismatch between the generative model of data and the learning model. To enhance the learning model to account for the presence of multiple sub-populations and labeling rules, a new strategy is developed where multiple neural networks are trained with the exchange of penalty.
+. Extensive experiments are conducted.

Weakness:

-. The proposed data generation framework is too simple. The framework is linear and without any uncertainty.
-. This paper presents many analytical and experimental results which affect the bias in the classification tasks. It would be more convincing to discuss how to identify these sources of bias from a real dataset, i.e., define metrics for those sources of bias.


**Summary Of The Paper:**

This manuscript introduces a framework to model data generation where the many bias-inducing factors are allowed for an exploration of the bias inheritance mechanism. Through the parametric framework, the authors analyze the data imbalance problem, investigate the various sources of bias, and propose a novel mitigation strategy based on matched inference approach, consisting of the introduction of coupled learning models. Experiments show that the coupled strategy can strike superior fairness-accuracy trade-offs.

**Summary Of The Review:**

This paper presents a generation prototype where the key factors of bias are analyzed theoretically and experimentally. Then a mitigation approach using coupled networks is proposed.

The framework illustrates the factors of bias but lacks quantitative measurement. The concepts of those factors are helpful for understanding bias in the data generation process but have some limitations to measuring bias in real data whose generation process is unavailable. In addition, the prototype is too simple compared with a real dataset.

---

### Official Review · Reviewer_jtpj · 2022-10-25

**Confidence:** 3
**Correctness:** 3
**Technical Novelty And Significance:** 2
**Empirical Novelty And Significance:** 2
**Recommendation:** 3

**Clarity, Quality, Novelty And Reproducibility:**

The paper is not well-written. All indices for appendices are missing.

Also, a lot of notations are used without definition.

The definition of Disparate impact (DI)  should be added to make the paper more self-contained.

In equation (2), $T_{c^\mu}$ is not defined, maybe it should be $W_T$.

Some notations are with \tilde, and some are not, which is hard for me to understand their differences.

In equation (3), $\mathbf{W}$ is used without definition. Are they corresponding to the weights $w$ trained using (2)? What makes it even more confusing, $w$ is reused in equation (8) as the optimizing variable.

Also, how to interpret $\delta q$ in equation (4)?


**Strength And Weaknesses:**

Strength:
It is nice to have a synthetic framework where all the parameters are under our control to help us understand the unfairness issue in real data. Also, it is good to see that the results are obtained in the over-parametrized regime, where $n,d \to \infty$, and $n/d<1$.

Weaknesses:
However, the insights obtained from this model seem to be very vague. It is unclear to me how the figures and remarks provided in section 3 can be useful in practice. A better way to present the result might be to focus on the trade-off between the model performance and fairness measure and show how the parameter of the T-M model will influence the trade-off.

Moreover, the proposed coupled networks are only discussed on the last page using one figure. If the authors want to emphasize such a contribution in designing a new algorithm by studying the proposed T-M model, more detailed discussions are needed, and significant rewriting is recommended.

For Figure 2, I cannot understand why a smaller value of $m_{+/-}$ leads to better fair performance. To my understanding, the smaller $m_{+/-}$ is, the more correlation between the label and the group, which should lead to worse fairness.

Figure 4, it is said in the caption that the first plot is about the accuracy gap, but the y axile of the plot is Disparate impact.

Figure 5, third column, what is "predicted parity 10"?


**Summary Of The Paper:**

This paper proposes a high-dimension model by combining the Gaussian Mixture and Teacher-Student setups to understand the unfairness issue of the real dataset. Through the tools of statistical physics, analytical characterizations of the typical properties of learning models trained in this synthetic “T-M” framework are provided. Despite the simplicity of the data model, the proposed framework is able to reflect some unfair behaviors observed on real-world datasets. Moreover, a basic loss-reweighing scheme is re-examined in this framework, which allows for an implicit minimization of different unfairness metrics and quantifies the incompatibilities between some existing fairness criteria.

**Summary Of The Review:**

There are some interesting aspects of the paper, but it cannot be published in its current form.

---

### Official Review · Reviewer_9w79 · 2022-10-28

**Confidence:** 3
**Correctness:** 2
**Technical Novelty And Significance:** 2
**Empirical Novelty And Significance:** 2
**Recommendation:** 3

**Clarity, Quality, Novelty And Reproducibility:**

Overall, I personally find the paper not easy to follow. There are multiple terms that appear in the paper as a key word (although come with references occasionally), e.g., geometrical properties, ML fairness phenomenology, statistical physics, structural feature, multi-task learning, etc. I think it would be better if authors can clarify how those terms/tools fit in the analysis, what is the goal, and what are the takeaway messages.

**Strength And Weaknesses:**

## Strength

The strength of the paper lies in the effort to consider the problem of ML bias under certain data modeling. The paper focuses on "data geometry" and presents analytical results accordingly.

## Weakness

The weakness of the paper comes from the difficulty to parse the results and therefore, to understand the goal and contribution.

### 1. what exactly does the word "bias" mean

While I understand the fact that "bias" is a heavily overloaded term in the literature, I think it would be better if the paper can make the term more precise and informative. The definition of "bias" is not presented until Page 4, and the biases are described in the form of a remark. I understand that (please correct me if I were wrong) the authors may want to have some generality in the discussion, but I do think it is necessary to make sure readers do not get lost, especially when the technical setup (the T-M model, the notations, the loss functions, etc.) is presented in full detail before readers even know what kind of bias the paper is trying to deal with.

### 2. the bias of interest

As a follow-up on 1., it seems that Sections 3 and 4 further narrow down the kind of bias to _disparate impact_, and consider a subset of ones in Remark 1. This way, I am not sure how to parse Remark 1, and the analytical results in Section 2. On one hand, the paper presents one specific type of data modeling assumption (which, may not corresponds to reality, as pointed out by authors), and the listed biases are specifically tailored to such data modeling (e.g., in Remark 1, all listed biases can only be instantiated after laying out the T-M modeling), and so are the two analytical results in Section 2. On the other, the analyses on source investigation (Section 3) and mitigation strategy (Section 4) are conducted after "choos[ing] a metric of fairness" (here is DI). This inconsistency of fairness notions is very confusing. What is the intended takeaway message? Further clarifications would be very helpful.

### 3. question on the claimed contribution

In Section 1, the paper claims that the work "allow an exact quantification of the intrinsic trade-offs between [previous group-level fairness notions]". I am having some difficulties connecting the presented analyses to this goal. In particular, in light of the concerns in 1. and 2., it would be very helpful if authors can share some insights on how tools of statistical physics and data geometry can shed light on bias quantification and mitigation. For example, analytical result 1 is a "non-rigorous yet exact replica method from statistical physics", what should be the implication of this result?

### 4. the referenced literature

In Section 1, to the best of my knowledge, the fairness notion presented by Dwork et al. (2012) is not group-level fairness.

**Summary Of The Paper:**

The paper proposes to investigate ML bias through the "data geometry". In particular, the authors introduce a Teacher-Mixture (T-M) model, which is a combination of Gaussian Model and Teacher-Student model, and consider potential biases under this specific type of data modeling. Analytical and empirical results are provided.

**Summary Of The Review:**

The paper proposes to investigate ML bias through the data geometry. The authors introduce T-M model, and investigate bias under this specific type of data modeling. There are worries about the clarity of the paper and the intended theoretical and technical contribution.

====Post Rebuttal====

I acknowledge that I have read reviewers' comments, authors' responses, and have incorporated them in evaluation.

---

### Official Review · Reviewer_CwVZ · 2022-10-29

**Confidence:** 3
**Correctness:** 4
**Technical Novelty And Significance:** 3
**Empirical Novelty And Significance:** 3
**Recommendation:** 5

**Clarity, Quality, Novelty And Reproducibility:**

The paper is well written and clearly presented. The analytical solution to this syntethic data generation problem is, I believe, novel

**Strength And Weaknesses:**

Strength:

The paper is well written, the data model is simple and understandable, and the results are clearly expressed in terms of key parameters such as covariate variance, relative group ratios, label frequencies, feature-to-sample ratios, and similarities in the labeling rule.

Some of these parameters lend themselves to quality discussions on existing approaches such as group re-weighting techniques, and on positive data transfer as a function of data scarcity and label rule similarity.

Weaknesses:

Though the approach is novel, I do not think the results highlight any truly novel aspect of bias. The label rebalancing literature already has results on the analytical solution of, for example, minimax cross-entropy solutions to the reweighted problem that similarly highlight better transfer for similar posterior prediction distributions p(Y|X, G) without specific assumptions on the data model (Although here the results are more concisely grounded in terms of data scarcity as well, which is a nice addition). The discussion of Coupled Networks is interesting, but seems to be only briefly mentioned and not entirely grounded in the theory and results shown in the paper (e.g., why elastic penalties? how well does this work in practice?, how does it compare to existing methods that have train-time access to group membership?)

**Summary Of The Paper:**

The paper analyzes common fairness metrics achieved by empirical risk minimization on a synthetic data generation model. The work leverages existing physics results to fully characterize solutions for two-group, binary label scenarios where the group-conditional covariate distribution follows a simple Gaussian distribution, and the group-conditional labels are determined by a simple hyperplane over the covariates.

The analytical solution to this simple model enables the authors to characterize (in particular) disparate impact and positive transfer as a function of key model parameters

**Summary Of The Review:**

The paper uses techniques from statistical physics to analytically compute the ERM solution to a binary classification problem with two demographic groups. The analytical solution is used to discuss aspects of transfer learning and sample reweighting. Though the model and results themselves seem novel, I don't see particularly impactful insights that arise from the paper's analysis beyond the scope of the particular data model.

---

### Decision · Program_Chairs · 2023-01-20

**Decision:**

Reject

**Justification For Why Not Higher Score:**

Reviewers found that it was difficult to parse the results in the paper and therefore, to understand the goal and contribution; moreover, the insights obtained from this model seem to be very vague.

**Justification For Why Not Lower Score:**

It is nice that the paper considers the problem of ML bias under certain data modeling, with a simple and intuitive framework.

**Metareview: Summary, Strengths And Weaknesses:**

This paper proposes Teacher-Mixture (T-M) model, as a high-dimension model by combining the Gaussian Mixture and Teacher-Student setups to understand the unfairness issue of the real dataset. It is nice that the paper considers the problem of ML bias under certain data modeling, with a simple and intuitive framework. However, it is difficult to parse the results in the paper and therefore, to understand the goal and contribution; moreover, the insights obtained from this model seem to be very vague.